# Determination of the trend of incidence of cutaneous leishmaniasis in Kerman province 2014-2020 and forecasting until 2023. A time series study

**Parya Jangipour Afshar**[1], **Abbas Bahrampour**[2], **Armita Shahesmaeili**[3]\*

**1** Student Research Committee, Kerman University of Medical Sciences, Kerman, Iran, Department of Biostatistics and Epidemiology, Faculty of Public Health, Kerman University of Medical Sciences, Kerman, Iran, **2** Modeling in Health Research Center, Institute for Futures Studies in Health, Department of Biostatistics and Epidemiology, Faculty of Health, Kerman University of Medical Sciences, Kerman Iran, **3** HIV/STI Surveillance Research Center, and WHO Collaborating center for HIV surveillance Institute for Futures Studies in Health, Kerman University of Medical Sciences, Kerman, Iran

\* armita.shahesmaeili@gmail.com

**Data Availability Statement:** All relevant data are within the manuscript and its Supporting Information files.

## Abstract

### Introduction

Cutaneous leishmaniasis (CL) is currently a health problem in several parts of Iran, particularly Kerman. This study was conducted to determine the incidence and trend of CL in Kerman during 2014–2020 and its forecast up to 2023. The effects of meteorological variables on incidence was also evaluated.

### Materials and methods

4993 definite cases of CL recorded from January 2014 to December 2020 by the Vice-Chancellor for Health at Kerman University of Medical Sciences were entered. Meteorological variables were obtained from the national meteorological site. The time series SARIMA methods were used to evaluate the effects of meteorological variables on CL.

### Results

Monthly rainfall at the lag 0 ($\beta$ = -0.507, 95% confidence interval:-0.955,-0.058) and monthly sunny hours at the lag 0 ($\beta$ = -0.214, 95% confidence interval:-0.308,-0.119) negatively associated with the incidence of CL. Based on the Akaike information criterion (AIC) the multivariable model (AIC = 613) was more suitable than univariable model (AIC = 690.66) to estimate the trend and forecast the incidence up to 36 months.

### Conclusion

The decreasing pattern of CL in Kerman province highlights the success of preventive, diagnostic and therapeutic interventions during the recent years. However, due to endemicity of disease, extension and continuation of such interventions especially before and during the time periods with higher incidence is essential.

**Funding:** The authors received no specific funding for this work.

**Competing interests:** The authors have declared that no competing interests exist.

## Author summary

Cutaneous leishmaniasis (CL) is one of the most prevalent tropical diseases and the most common form of leishmaniasis, which is found in different regions. Due to different geographical climates, the transmission pattern and the impact of meteorological variables on CL is different. In this study we evaluated the incidence and trend of CL during 2014–2020 and its forecast up to 2023 in Kerman province, Iran. In addition, the impact of meteorological variables on its incidence was assessed. Our finding showed a decreasing trend of CL during the studied years. There was a negative association between CL and sunny hours per day and rainfall at lag 0.

## Introduction

Cutaneous leishmaniasis (CL) is a type of leishmaniasis transmitted to mammals by the bite of a female sand fly [1]. According to the World Health Organization (WHO), CL is one of the six major tropical diseases [2]. Different reports showed that the incidence of leishmaniasis is increasing [3,4].

In 2012, WHO reported that the highest rates of the disease were in 10 countries of Afghanistan, Algeria, Colombia, Brazil, Iran, Syria, Ethiopia, North Sudan, Costa Rica and Peru which accounted for about 70–75% of cases [5]. According to global estimates in 2015, nearly 4 million people suffered from leishmaniasis, equating to approximately 46,000 years of healthy life lost due to disability (YLDs) which was corresponding to 27.3% increase in incidence and 25.5% in YLDs compared to 2005 [6]. Annually, around 20,000 new cases of leishmaniasis, both rural and urban, are being reported from different parts of Iran [7]. In this country, the number of new cases per 100,000 population increased from 50 in 1977 to 250 in 2015 with the dominance of men. Furthermore, the burden of CL raised from 1.18 to 5.7 DALYs per 100,000 population during these years [8].

Kerman province, located in the Southeast of Iran, and particularly the city of Bam, is one of the endemic areas for CL. In a review conducted in 2015 in Kerman province, cities of Bam and Kerman were the most infected areas, with an incidence of 63.6% and 24.7%, respectively [9]. Although CL seems to be a continuing health problem in this area, no study has been conducted to investigate the current and future trends of disease in Kerman. Therefore, the aim of the present study was to investigate the trend of CL during 2014 to 2020 and its forecast up to the year 2023.Effect of meteorological variables on disease incidence was also evaluated.

## Methods

### Ethical statement

The study protocol was approved by the Graduate Studies Council and Ethics Committee of Kerman University of Medical Sciences (Ethics code: IR.KMU.REC.1399.698).

### Area of study

Kerman Province, with its hot and dry climates, is located in the southeast of Iran. It covers an area of 183193 $km^2$ and had a population of approximately 3.2 million people in 2016, it accounts for nearly 11 percent of the land area and 3.5 percent of the population of Iran. It is located between 30 17′ 24″/E and 57 3′ 36″/N. The average annual temperature and rainfall are 15.8˚C and 132.4 mm, respectively. (Fig 1)

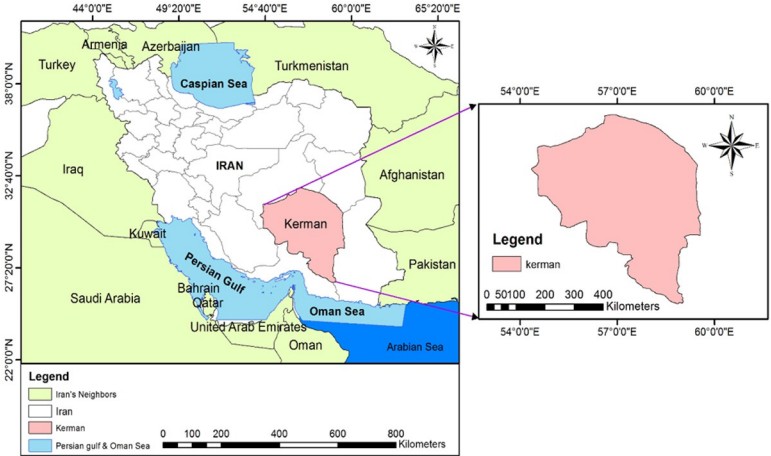

**Fig 1. Geographical location of the study area, the Kerman province, Iran.** The main data has been prepared in shapefile (shp) format from https://www.diva-gis.org/datadown.

## Data collection

The monthly number of confirmed cases of CL from January 2014 to December 2020 was obtained from vice chancellor of health affiliated to four medical universities throughout the province. Meteorological information, including monthly synoptic information such as monthly average temperature (˚C), average maximum temperature (˚C), average minimum temperature (˚C), monthly average rainfall per 24 hours (mm), average sunny hours per day and average relative humidity () in each month was extracted from the website of Meteorological Office (www.farsmet.ir) during the study period.

## Statistical analysis

To model and predict the number of CL cases, SARIMA model (p, d, q) (P, D, Q) was along with the Box-Jenkins method used. p was the number of autoregressive; d was the number of model differentiation; q was the number of regressive moving average in non-seasonal mode; P was the number of seasonal autoregressive; D was the number of model differentiation in seasonal mode; Q was the number of the moving regression in seasonal model. The seasonal period(s) considered 12 months. The following steps were taken to fit the model.

As the main assumption in time-series analysis is stationary (independence of series from time), we applied the Box-Cox and Dickey Fuller tests to evaluate the variance and mean stability of the model over time; The null hypothesis in these tests is that the mean and variance of series are stable over time and there is no seasonal trend in series. In the absence of variance stability, the appropriate conversion was performed according to the Box-Cox test value. This procedure balance the seasonal fluctuations and random variation across the series. When the variance stability was obtained, the mean stability was investigated. In the case of no mean stability, first-degree difference (D = 1) was used and stability was obtained with one differential degree. Moreover, in the case of a seasonal trend, the first-degree seasonal difference (D = 1) was applied with the 12th period. In the next step, the autocorrelation function (ACF) and partial autocorrelation function (PACF) were used to determine the AR (p, P) and MA (q, Q) parameters. These auto correlation functions are correlation between variable's current value and its past value and show which past series values are most useful in predicting future values. In other word, ACF at lag k indicate the correlation between series values that are k intervals apart. However, PACF at lag k indicate the correlation between series values that are k intervals

apart, accounting for the values of the intervals between. We plotted ACF and PACF to present the finding. In these plots the *x* axis represents the lag at which the autocorrelation is computed; the *y* axis indicates the value of the correlation which ranges from -1 to 1. A positive correlation indicates that large current values correspond with large values at the specified lag; a negative correlation indicates that large current values correspond with small values at the specified lag. Furthermore, we used Akaike information criterion (AIC) and Bayesian information criterion (BIC) indices to compare the various SARIMA models. In these indices, a lower value demonstrates a better model. The likelihood ratio test was also applied to select the best model, a higher value indicated a better fit. In order to evaluate the final model fit, the normality of the residuals was evaluated using histogram chart and Shapiro-Wilk tests, the p-value greater than 0.05 indicated normality of residuals. The Ljung-box (Q) test was also used to investigate whether the residuals had white noise (mean = zero and constant variance) or not, the p-value greater than 0.05 indicated white noise of residuals.

To improve prediction, meteorological variables (maximum monthly temperature, minimum monthly temperature, temperature, monthly 24-hour rainfall, average sunny hours per day and average relative humidity) were entered in to the model; to eliminate the correlation and seasonal trend of each series, the pre-whitening method was used. A SARIMA model was separately obtained for each series. The variance inflation factor (VIF) was used to determine the significant correlation (collinearity) between meteorological variables, if any variable have VIF$\geq$10 that means high collinearity. Only variables with a VIF <10 were entered to the final model. To identify the appropriate time lags of the independent variables, the cross-correlation coefficients (CCF) chart measuring the effect of each independent variable on the dependent variable was used. After the appropriate lags were identified, each independent variable was entered into the ARIMAX model with its own lags. Finally, the AIC and BIC were calculated to find the most suitable model. The correlation between residuals and white noise error was checked for the final model and the predictive values of the final model for 36 months later were presented. In this study, the "tseries", "forcast" and "TAS" packages in the R software (version 4.1.0) were used to analyze data. P-values less than 0.05 were considered significant.

## Results

### Descriptive analysis

A total of 4993 cases were entered into the study. Our findings indicate a decreasing trend of CL from 2014 (951 cases) to 2020 (430 cases), with the highest incidence occurring at the beginning of spring and end of autumn (Fig 2). The characteristics of meteorological variables are also given in Table 1.

### Univariable model

Based on AIC, BIC and value of Shapiro-Wilk test, ARIMA (0,1,2) (2,0,0) $_{12}$ was the best model for the series of CL cases (Fig 3). Moreover, Ljung-box (Q) test (P = 0.26) revealed that the model residuals were white noise, indicating that the model was suitable.

### Multivariable model

Based on stability analysis and VIF, four meteorological variables including average temperature, monthly 24-hour rainfall, average sunny hours per day and average relative humidity entered in to the final ARIMAX (multivariable) model (Table 2), and their effective lags were determined by using CCF diagram.The ARIMA (0,1,1) (0,0,1)$_{12}$ was chosen as the best model. In this model, "rainfall" ($\beta$ = -0.507) and "sunny hours" ($\beta$ = -0.214), both at the lag of 0 were

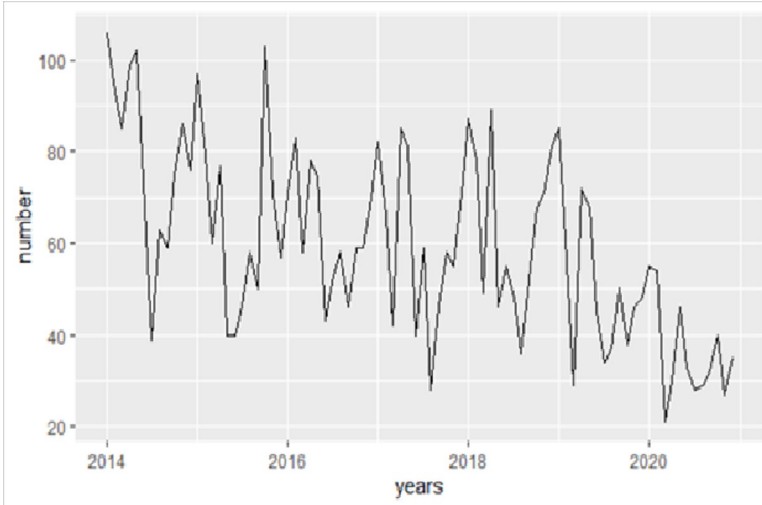

**Fig 2. The trend of cutaneous CL in Kerman province, Iran during 2014–2020.**

negatively associated with the incidence of CL. The coefficients and statistics of the model are given in Table 3. Shapiro-Wilk test (P = 0.23) and Ljung-box (Q) (p = 0.302) confirm the normality and white noise of residuals, respectively. The AIC and BIC confirm that this model is more suitable than the univariable model for predicting the number of CL cases up to the next 36 months. The model based estimates series and forecasted number of cases are depicted in Fig 4.

## Discussion

In present work, we studied the trend of cutaneous leishmaniasis from 2014 to 2020 in Kerman province, an endemic area of leishmaniasis in Iran and evaluated the relation between metrological factors and disease incidence. Our findings indicate an overall seasonal decreasing trend of CL incidence within the study period which is in line with the decreasing pattern of disease in different parts of Iran [10–12]. Global warming, reduced rainfall, extension of preventive interventions, increased access to diagnostic and treatment facilities may all be contributing to this reduction. Furthermore, as we showed, the highest incidence of CL was seen at the beginning of spring and end of autumn. Several studies have shown the seasonal pattern of CL. In line with our findings, a study conducted by Wasserberg G et al. in 2003 showed a higher incidence during the autumn and spring [13]. Similarly, in a study conducted in 2013 in Isfahan, Iran, an endemic city for leishmaniasis, the most cases of infection reported to occur during the summer and autumn [2]. However, in a study conducted in 2003 in Pakistan [14], as well as two separate studies conducted during 2007–2016 in west of Iran and in 2016 in southwest of Iran [15,16] a peak was shown during the winter. Another study conducted during 2009–2016 in central region of Iran indicate the higher incidence in the autumn and winter [12]. The common finding of most of these studies with our study is that the higher incidence of infection was seen in humid seasons including spring, autumn and winter where increased humidity may provide a suitable condition for the growth and reproduction of mosquitoes. However the amount of moisture can vary in each season depending on the geographical location. This phenomenon explains the difference between findings of studies. For example, in Kerman province where the present study was done, the highest level of moisture

**Table 1. Descriptive characteristics of meteorological variables in Kerman province, Iran during 2014–2020.**

| Year | Variables | Median | Mean | Standard error | Minimum | Maximum |
|------|-----------|--------|------|----------------|---------|---------|
| 2014 | Maximum monthly temperature (˚C) | 29.29 | 27.68 | 9.63 | 13.65 | 39.63 |
| | Minimum monthly temperature(˚C) | 14.64 | 13.08 | 8.66 | 1.46 | 24.16 |
| | Monthly Rainfall(mm) | 4.44 | 11.23 | 15.04 | 0.01 | 46.95 |
| | Average Relative humidity () | 29.62 | 31.35 | 13.94 | 15.26 | 57.42 |
| | Monthly average of sunny hours per days | 275.11 | 276.960 | 48.66 | 208.34 | 349.02 |
| | Average temperature(˚C) | 22.06 | 20.54 | 9.56 | 7.25 | 32.57 |
| 2015 | Maximum monthly temperature (˚C) | 30.92 | 28.46 | 8.44 | 16.39 | 38.81 |
| | Minimum monthly temperature(˚C) | 15.83 | 13.95 | 7.63 | 2.67 | 24.10 |
| | Monthly Rainfall(mm) | 6.19 | 11.28 | 12.35 | 0.04 | 39.43 |
| | Average Relative humidity () | 25.25 | 30.22 | 13.29 | 15.33 | 47.63 |
| | Monthly average of sunny hours per days | 272.75 | 272.31 | 51.11 | 191.46 | 345.10 |
| | Average temperature(˚C) | 23.61 | 21.34 | 8.52 | 345.10 | 31.84 |
| 2016 | Maximum monthly temperature (˚C) | 28.83 | 29.22 | 7.95 | 18.15 | 40.34 |
| | Minimum monthly temperature(˚C) | 13.67 | 14.03 | 7.65 | 3.62 | 25.55 |
| | Monthly Rainfall(mm) | 2.19 | 4.21 | 5.37 | 0.02 | 16.79 |
| | Average Relative humidity () | 26.33 | 26.35 | 9.35 | 13.03 | 39.72 |
| | Monthly average of sunny hours per days | 273.02 | 281.94 | 44.64 | 225.28 | 357.20 |
| | Average temperature(˚C) | 21.46 | 21.82 | 8.19 | 357.20 | 33.42 |
| 2017 | Maximum monthly temperature (˚C) | 30.94 | 28.58 | 8.37 | 16.32 | 40.09 |
| | Minimum monthly temperature(˚C) | 14.79 | 13.63 | 7.74 | 2.59 | 23.54 |
| | Monthly Rainfall(mm) | 2.23 | 10.21 | 19.86 | 0.01 | 65.27 |
| | Average Relative humidity () | 25.39 | 27.50 | 12.73 | 12.60 | 54.90 |
| | Monthly average of sunny hours per days | 290.280 | 281.22 | 61.13 | 168.78 | 358.36 |
| | Average temperature(˚C) | 23.072 | 21.31 | 8.66 | 10.02 | 32.72 |
| 2018 | Maximum monthly temperature (˚C) | 28.42 | 29.36 | 7.73 | 19.02 | 39.78 |
| | Minimum monthly temperature(˚C) | 14.67 | 14.74 | 7.28 | 3.54 | 24.26 |
| | Monthly Rainfall(mm) | 2.56 | 5.25 | 6.70 | 0.01 | 19.94 |
| | Average Relative humidity () | 29.63 | 27.46 | 10.50 | 14.19 | 43.43 |
| | Monthly average of sunny hours per days | 270.48 | 277.64 | 49.06 | 196.70 | 348.58 |
| | Average temperature(˚C) | 21.43 | 22.23 | 7.91 | 11.25 | 32.67 |
| 2019 | Maximum monthly temperature (˚C) | 28.23 | 28.05 | 9.18 | 16.76 | 40.66 |
| | Minimum monthly temperature(˚C) | 14.60 | 14.06 | 8.20 | 4.38 | 25.78 |
| | Monthly Rainfall(mm) | 2.26 | 12.49 | 14.44 | 0.00 | 35.04 |
| | Average Relative humidity () | 31.21 | 30.55 | 12.34 | 14.74 | 47.57 |
| | Monthly average of sunny hours per days | 257.18 | 269.42 | 51.61 | 195.66 | 344.88 |
| | Average temperature(˚C) | 21.38 | 21.15 | 9.05 | 10.65 | 33.79 |
| 2020 | Maximum monthly temperature (˚C) | 27.09 | 27.66 | 9.27 | 13.52 | 39.91 |
| | Minimum monthly temperature(˚C) | 12.53 | 13.62 | 8.22 | 2.74 | 25.39 |
| | Monthly Rainfall(mm) | 5.09 | 15.28 | 20.33 | 0.00 | 55.67 |
| | Average Relative humidity () | 29.26 | 31.79 | 14.55 | 16.02 | 57.96 |
| | Monthly average of sunny hours per days | 268.55 | 268.68 | 50.90 | 199.18 | 341.02 |
| | Average temperature(˚C) | 19.80 | 20.75 | 9.08 | 7.78 | 32.64 |

usually is seen at the end of autumn and at the beginning of winter and spring when the weather is more humid.

Based on the multivariable ARIMA $(0,1,1)$ $(0,0,1)_{12}$ model, we showed a negative association between CL and sunny hours per day and rainfall at lag 0. The association between sunny

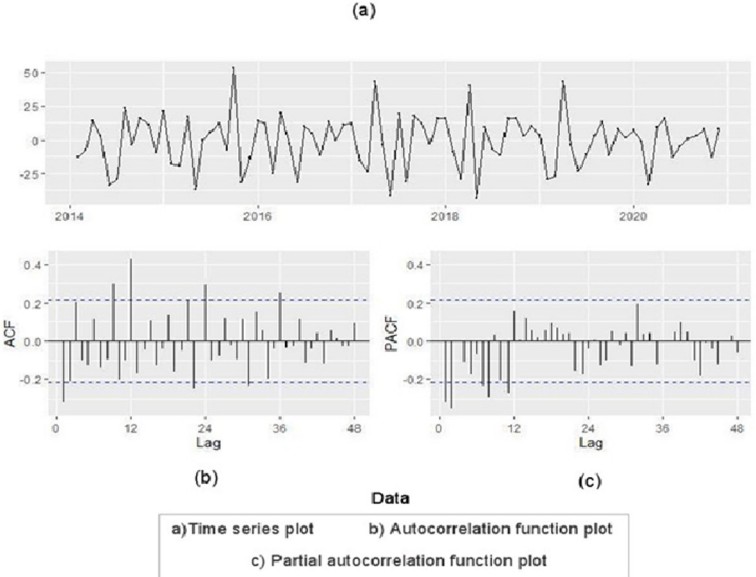

**Fig 3. Autocorrelation, partial autocorrelation function and time series plots of leishmaniasis in Kerman, Iran during 2014–2020 after one time differentiation.** The *x* axis represents the lag at which the autocorrelation is computed; the *y* axis indicates the value of the autocorrelation.

hours and incidence of leishmanial has been assessed in various studies. Similar to our study, Rahmanian, V et al. in their study in Isfahan, Iran, showed a negative association between sunny hours and the incidence of CL [10]. However, in another study conducted in 2021 in Iran, the average temperature and sunny hours positively associated with the incidence of CL [17]. Although it is expected that increase in sunny hours, and consequently temperature provide a suitable condition for vector (sandfly) and activity of reservoir, some recent studies argue that this effect may be non-linear. Adegboye, M et al. in their study in Afghanistan showed that temperature may have non-linear effect on the incidence of CL [18].This means that sandflies need an optimum temperature to be active and both downward and upward deviance from the optimum temperature may affect their activity. Therefore, the nonlinear effect of sunlight and temperature on disease incidence may explain the discrepancy between the findings of different studies. Furthermore, while previous studies indicate both positive and negative effect of temperature on disease incidence [18–20], we didn't find any association between mean temperature and incidence of disease. One explanation could be that Kerman province has a desert climate. Although in such climate the difference between day and night temperatures is high, the range of change in average daily temperature throughout the year is relatively narrow. In the absence of large variations, the effect of temperature on CL couldn't

**Table 2. Optimal models and parameter values for meteorological variables in Kerman province, Iran during 2014–2020.**

| Variables | Model | SMA1 | MA1 | SAR1 | SAR2 | AR1 | AIC |
|---|---|---|---|---|---|---|---|
| Monthly Rainfall (mm) | ARIMA $(0,0,0)(0,1,1)_{12}$ | -0.792 | NA | NA | NA | NA | 260 |
| Average Relative humidity () | ARIMA $(0,0,1)(0,1,1)_{12}$ | -0.740 | 0.346 | NA | NA | NA | -31.06 |
| Monthly Sunny hours per days | ARIMA $(0,0,0)(2,1,0)_{12}$ | NA | NA | -0.636 | -0.218 | NA | 670 |
| Average temperature (˚C) | ARIMA $(1,0,0)(2,1,0)_{12}$ | NA | NA | -0.925 | -0.471 | 0.228 | 293 |

AR: Auto-regressive, MA: Moving average, SAR: Seasonal auto-regressive, SMA: Seasonal moving average, NA: not applicable, AIC: Akaike Information Criterion

**Table 3. Coefficients and statistics of multivariable ARIMA (0,1,1) (0,0,1)$_{12}$ time series model of cutaneous leishmaniasis during 2014–2020 in Kerman province, Iran.**

| Variable | Lag | Estimate | Std.Error | Z-value | P-value |
|---|---|---|---|---|---|
| Constant | | 111.882 | 20.76 | 5.381 | 0.000 |
| MA1 | | -0.859 | 0.053 | -15.962 | 0.000 |
| SMA1 | | 0.454 | 0.146 | 3.104 | 0.000 |
| Monthly Rainfall(mm) | Lag 0 | -0.507 | 0.229 | -2.214 | 0.030 |
| Monthly sunny hours | Lag 0 | -0.214 | 0.048 | -4.414 | 0.000 |
| Average Relative humidity () | Lag 0 | -0.095 | 0.679 | -0.141 | 0.888 |
| Average temperature (˚C) | Lag 0 | -0.523 | 0.725 | -0.721 | 0.472 |
| Monthly sunny hours | Lag 8 | 0.011 | 0.040 | 0.286 | 0.774 |
| Monthly Rainfall (mm) | Lag 2 | 0.123 | 0.166 | 0.742 | 0.460 |
| Monthly Rainfall (mm) | Lag 5 | -0.173 | 0.167 | -1.034 | 0.340 |
| Average Relative humidity () | Lag 9 | 0.037 | 0.291 | 0.129 | 0.897 |

AIC = 613 AICc = 614.71 BIC = 627.38

AIC: Akaike Information Criterion, AICc: Akaike Information Criterion corrected

BIC: Bayesian Information Criterion, MA: Moving average, SMA: Seasonal moving average

be addressed well. Additionally a wide range of environmental, structural and biologic factors may mediate the effect of sunlight and temperature on dieses. For example in Afghanistan, an inverse association between temperature and disease incidence was projected to increased indoor activity of population in cold seasons and endophagic character of the sandflies. However in other studies, the positive association between temperature and CL incidence has been justified by increased activity of sandflies and vectors in warmer temperatures [19,20]. It has been shown depending on the type of vector, temperature may disproportionally affect the life-cycle of leishmania [21]. All of these may explain contradictory effect of sunlight and temperature on disease incidence seen in various studies.

Surprisingly, we saw an inverse association between amount of rainfall and incidence of CL. While a couple of studies including studies conducted by Yamada K et al. in Panamá

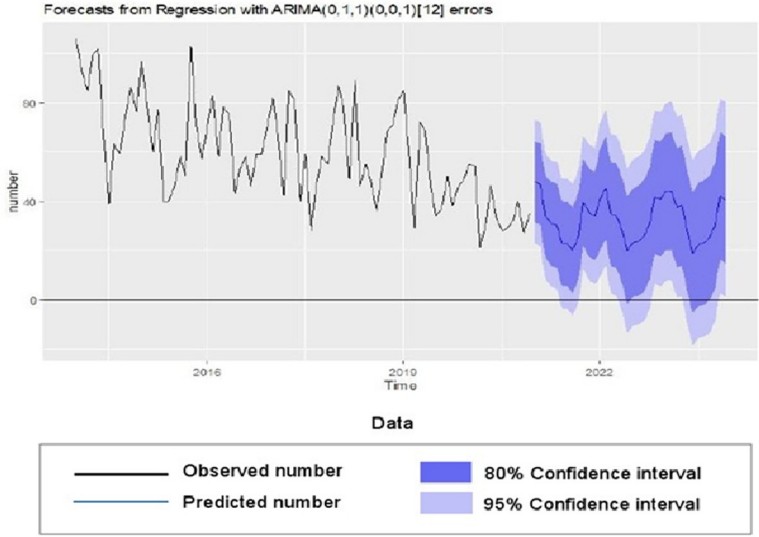

**Fig 4. Forecasting the number of cutaneous leishmaniasis in Kerman province, Iran up to 2023.**

(2016) [22], Sharafi M et al. in southern Fars, Iran (2016) [23] and Talmoudi, K et al. in Tunisia (2017) [24] support the most common idea that increase in rainfall and consecutively humidity may increase the incidence of disease, there are some studies that have come to the opposite conclusion. For example, a study conducted in 2017 in eastern Fars, Iran, showed a negative association between rainfall and CL incidence, but a positive association between relative humidity and the disease [11]. Another study conducted in French Guiana, in 2013 showed a positive association between mean temperature and disease incidence while a negative association between rainfall and incidence of disease [25]. Similarly, two separate studies conducted in Sri Lanka during 2009–2016 and Central Tunisia in 2017 showed that monthly average temperatures has a positive correlation with the incidence of leishmaniasis; while monthly average rainfall has a negative association with the incidence of disease [26,27]. In a study conducted in 2016 in Kenya, soil temperature, rainfall and relative humidity were negatively correlated with abundance of sandflies [28]. It seems that the effect of rainfall on the incidence of CL being mediated by other environmental factors. For example in areas with mild climate where the moisture created after the rain may remain for a longer period, rainfall could facilitate the spawning of insects and increase the egg survival. However, in areas like Kerman province, where the weather is mostly hot and dry and moisture doesn't remain for a long time, rainfall could bring about decrease in temperature and reduction in sunny hours which prohibits the activity of insects [25]. On the other hand, although the average rainfall in Kerman is much lower than the national average, the rains are occasionally heavy and torrential. Available literature shows that heavy rains can negatively affect the sandflies by restricting its flight and killing immature insects [29]

Our work provides a valuable insight for policy makers regarding the current situation of CL in an endemic area of Iran. Furthermore, in contrast to studies that applied univariable models of climate factors, we used multivariable SARIMA model to evaluate the effect of metrological factors on the incidence of disease. While our finding support the role of environmental factors in disease incidence, due to some limitations, the findings should be interpreted with caution. First, because of under-reporting and under-diagnosis of cases with relatively mild symptoms, this study may under-estimate the true incidence. In addition, there would be the possibility of confounding by unmeasured confounders.Furthermore, we didn't study the non-linear effect of meteorological factors on incidence. Finally, this is an ecological study and therefore its findings couldn't be generalizable to the individual level.

## Conclusion

We showed a decreasing trend of cutaneous leishmaniasis in Kerman, Iran with a seasonal pattern at the end of autumn and beginning of spring. Despite the decreasing trend of disease, it is still considered as an endemic disease in Kerman province. Extension and continuation of preventive interventions, as well as improvements in diagnosis, care and treatment especially before and during the time periods with higher incidence is essential for control of disease and should be emphasized by policymakers in future planning.

## Supporting information

**S1 Data. This file consisting of monthly CL was recorded from January 2014 to December 2020 by the Vice-Chancellor for Health at Kerman University of Medical Sciences and Meteorological information of Kerman was extracted from the website of Meteorological Office (www.farsmet.ir) during the study period.**
(XLSX)

## Acknowledgments

We would like to thank all the people who helped us to perform this study, especially Staffs of the Vice-Chancellor for Health, Department of communicable diseases, Kerman University of Medical Science. Present article extracted from MSc thesis of PJ.

## Author Contributions

**Conceptualization:** Parya Jangipour Afshar, Armita Shahesmaeili.

**Data curation:** Parya Jangipour Afshar.

**Formal analysis:** Parya Jangipour Afshar, Abbas Bahrampour.

**Investigation:** Armita Shahesmaeili.

**Methodology:** Parya Jangipour Afshar, Armita Shahesmaeili.

**Project administration:** Armita Shahesmaeili.

**Software:** Parya Jangipour Afshar, Abbas Bahrampour.

**Supervision:** Abbas Bahrampour, Armita Shahesmaeili.

**Validation:** Abbas Bahrampour, Armita Shahesmaeili.

**Writing – original draft:** Parya Jangipour Afshar.

**Writing – review & editing:** Parya Jangipour Afshar, Abbas Bahrampour, Armita Shahesmaeili.

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
