## [Decision Letter · Decision Letter 0]

19 Nov 2021

Dear Dr. Shahesmaeili,

Thank you very much for submitting your manuscript "Determination of the trend of incidence of cutaneous leishmaniosis in Kerman province 2014-2020 and forecasting until 2023. A time series study" for consideration at PLOS Neglected Tropical Diseases. As with all papers reviewed by the journal, your manuscript was reviewed by members of the editorial board and by several independent reviewers. In light of the reviews (below this email), we would like to invite the resubmission of a significantly-revised version that takes into account the reviewers' comments. 

We cannot make any decision about publication until we have seen the revised manuscript and your response to the reviewers' comments. Your revised manuscript is also likely to be sent to reviewers for further evaluation.

Sincerely,

Johan Van Weyenbergh

Associate Editor

Kristien Verdonck

Deputy Editor

Reviewer's Responses to Questions

**Key Review Criteria Required for Acceptance?**

**Methods**

-Are the objectives of the study clearly articulated with a clear testable hypothesis stated?

-Is the study design appropriate to address the stated objectives?

-Is the population clearly described and appropriate for the hypothesis being tested?

-Is the sample size sufficient to ensure adequate power to address the hypothesis being tested?

-Were correct statistical analysis used to support conclusions?

-Are there concerns about ethical or regulatory requirements being met?

Reviewer #1: The study design, objectives and hypothesis are suitable and articulated. Aggregated data on cutaneous leishmaniasis incidence and meteorological information were used. The sample size is the cumulative number of cutaneous leishmaniasis cases between 2014 and 2020 in Kerman Province, Iran. The authors applied the correct statistical tests to support their conclusions. The authors describe that the study was approved by a local research ethics committee.

Reviewer #2: The manuscript is adequate in terms of its methodology. However, the exposition of the statistical methods needs improvement. The models being used should be described in more detail, at least up to the point that it is clear to the reader how to interpret the various model parameters that are mentioned.

**Results**

-Does the analysis presented match the analysis plan?

-Are the results clearly and completely presented?

-Are the figures (Tables, Images) of sufficient quality for clarity?

Reviewer #1: The analysis presented match the analysis plan. However, the results need to be completely presented. For example, Table 1 should present data by year and not only aggregated parameters for the entire period (2014-2020). The legend in figure 2 needs to be improved. It is also necessary to assign letters to its three images. The legend in table 3 needs to be improved. In figure 4, please indicate what are: black line, blue line, light and dark blue clouds.

Reviewer #2: The results are for the most part adequate, but please see my comments regarding "conclusions."

**Conclusions**

-Are the conclusions supported by the data presented?

-Are the limitations of analysis clearly described?

-Do the authors discuss how these data can be helpful to advance our understanding of the topic under study?

-Is public health relevance addressed?

Reviewer #1: The conclusions are supported by the data presented. But but the authors need to clearly define the main limitations of the study. The results need to be more widely discussed and compared to other endemic localities for cutaneous leishmaniasis.

Reviewer #2: I think the interpretation and discussion of the results could be more extensive and insightful. Also, the authors should clarify how their result of a negative association between rainfall at a lag of 0 and incidence of cutaneous leishmaniasis compares with findings in the literature that appear to suggest the opposite. Maybe this has to do with the lag times under consideration in other studies? In any case, the authors should clarify.

**Editorial and Data Presentation Modifications?**

Reviewer #1: the word leishmaniasis is misspelled several times throughout the text. Please correct. In lines 49-52, please use more recent references. In lines 62-63, the authors indicate that "the burden of the disease has not decreased significantly in recent years". The authors' data show an opposite trend for the Kerman province.

Reviewer #2: The writing in the paper can and should be improved. Please see my comments for some suggestions, but I recommend the authors go through the entire paper carefully to ensure that the wording is not awkward or unclear.

**Summary and General Comments**

Reviewer #1: Afshar et al. evaluated the incidence of cutaneous leishmaniasis in Kerman province (Iran) from 2014 to 2020 and estimated its incidence until 2023. Study strengths: The research question is original, relevant to the field and of interest to the journal's readers. The method used is robust, providing reliable and quality results. Study weaknesses: the results need to be more widely discussed and compared to other endemic localities for cutaneous leishmaniasis.

Reviewer #2: The manuscript needs improvement in several ways. I found the discussion and interpretation of the results to be rather brief, and a more thorough, insightful discussion would improve the manuscript. Also, please see my comments regarding the potential confusion related to the association between rainfall and CL incidence. Finally, the exposition of the statistical models that are used should be more detailed, and the writing in general can use some improvement.

Here are some specific comments/recommendations:

line 36: include confidence intervals

line 36: instead of writing the variables in quotation marks, describe them properly

line 43: change "a method" to "methods"

lines 53-55: change wording. Perhaps "WHO reported that the highest rates of the disease were in 10 countries which accounted for 70-75% of cases: ..."

line 70: Why only predict up to the year 2023? And what does the study hope to achieve by predicting the trend for the next two years? This needs to be clarified to provide proper motivation.

line 73: Use commas and decimal points as appropriate in the numbers. Also, I think the population

is more like 3.164718 million (it is missing a decimal point). Please correct.

line 74: It is a bit incomplete to note that it accounts for nearly 11 percent of the land area of Iran but not note what percent of the population it accounts for.

lines 80-81: change "obtained from vice chancellor of health" to "was obtained from the vice chancellor of health"

lines 90-96: Please provide more explanation about the statistical methods that you are using. Your explanation of the statistical models should be brief, but still detailed enough that the reader understands what the variables that you refer to correspond to.

line 131: Change "is also given" to "are also given"

line 178: Change wording of "an overall trend was decreasing" so that it is more clear.

line 179: Put comma after "As we showed"

lines 188-189: I am confused: if a reduction of rainfall is associated with a decrease in the incidence of CL, isn't that a positive association between rainfall and incidence? How does this compare to your findings of a negative association between rainfall at a lag of 0 and CL incidence?

lines 198-201: Does this not suggest a positive association between rainfall and CL incidence (which is in contrast to the negative association that you inferred between rainfall at a lag of 0 and CL incidence)?

lines 204-205: Again, a comment about reduced rainfall contributing to a decreasing trend of CL incidence. How do you reconcile this with your inference of rainfall (at the lag of 0) being negatively associated with CL incidence?
---

## [Decision Letter · Decision Letter 1]

11 Feb 2022

Dear Dr. Shaheshmaeili,

We are pleased to inform you that your manuscript 'Determination of the trend of incidence of cutaneous leishmaniasis in Kerman province 2014-2020 and forecasting until 2023. A time series study' has been provisionally accepted for publication in PLOS Neglected Tropical Diseases.

Best regards,

Johan Van Weyenbergh

Associate Editor

Kristien Verdonck

Deputy Editor

Reviewer's Responses to Questions

**Key Review Criteria Required for Acceptance?**

**Methods**

-Are the objectives of the study clearly articulated with a clear testable hypothesis stated?

-Is the study design appropriate to address the stated objectives?

-Is the population clearly described and appropriate for the hypothesis being tested?

-Is the sample size sufficient to ensure adequate power to address the hypothesis being tested?

-Were correct statistical analysis used to support conclusions?

-Are there concerns about ethical or regulatory requirements being met?

Reviewer #1: The study design, objectives and hypothesis are suitable and articulated. Aggregated data on cutaneous leishmaniasis incidence and meteorological information were used. The sample size is the cumulative number of cutaneous leishmaniasis cases between 2014 and 2020 in Kerman Province, Iran. The authors applied the correct statistical tests to support their conclusions. The authors describe that the study was approved by a local research ethics committee.

Reviewer #2: I think that the Methods are adequate.

**Results**

-Does the analysis presented match the analysis plan?

-Are the results clearly and completely presented?

-Are the figures (Tables, Images) of sufficient quality for clarity?

Reviewer #1: The analysis presented match the analysis plan. Now, the results are completely presented. The Table 1 now shows all the data (by year and not only aggregated parameters for the entire period).

As requested, Figure legends have been improved.

Reviewer #2: Yes, the Results are adequate.

**Conclusions**

-Are the conclusions supported by the data presented?

-Are the limitations of analysis clearly described?

-Do the authors discuss how these data can be helpful to advance our understanding of the topic under study?

-Is public health relevance addressed?

Reviewer #1: The results are more widely discussed and the main limitations are presented at the end of the discussion.

Reviewer #2: Yes, the Conclusions are adequate.

**Editorial and Data Presentation Modifications?**

Reviewer #1: The Authors have addressed all of my concerns

Reviewer #2: The authors should read over their manuscript one more time to ensure that they take care of all typographical, spelling and grammatical errors.

For example, in lines 236-238, they write:

"Furthermore, while previous studies indicate both positive and negative effect of temperature on disease incidence (17-19). We didn’t found any association between mean temperature and disease incidence. "

The first sentence ends prematurely, and the period should be replaced by a comma, combining the two fragments into a proper sentence. Also, "found" should be "find."

On line 249, they write "contraindicatory." I am guessing they meant to write "contradictory."

In numerous parts of the text, there are no spaces where there should be spaces, and sometimes there are spaces where there should be no space (for example, a space between a word and a period ending a sentence).

The issues I have identified in these comments do not constitute an exhaustive list. The authors should read the entire manuscript again very carefully and make sure that they identify and take care of all errors.

**Summary and General Comments**

Reviewer #1: The Authors have addressed all of my concerns

Reviewer #2: I think that the authors adequately addressed the substantial comments that I made in my review of their original submission. I recommend acceptance, but they should carefully correct all remaining errors in their manuscript before publication.

---

## [Editor Report · Acceptance letter]

26 Mar 2022

Dear Dr Shahesmaeili,

We are delighted to inform you that your manuscript, "Determination of the trend of incidence of cutaneous leishmaniasis in Kerman province 2014-2020 and forecasting until 2023. A time series study," has been formally accepted for publication in PLOS Neglected Tropical Diseases.

Best regards,

Shaden Kamhawi

co-Editor-in-Chief

Paul Brindley

co-Editor-in-Chief
